# Association of Slowly Digestible Starch Intake with Reduction of Postprandial Glycemic Response: An Update Meta-Analysis

**DOI:** 10.3390/foods12010089

**Published:** 2022-12-24

**Authors:** Yanli Wang, Xiao Zhou, Xuesong Xiang, Ming Miao

**Affiliations:** 1State Key Laboratory of Food Science & Technology, Jiangnan University, 1800 Lihu Avenue, Wuxi 214122, China; 2National Institute for Nutrition and Health, Chinese Center for Disease Control and Prevention, 27 Nanwei Road, Xicheng District, Beijing 100050, China

**Keywords:** slowly digestible starch, physiological function, extended glycaemic index, glycaemic profile, meta-analysis

## Abstract

Slowly digestible starch (SDS) has been shown to digest slowly throughout the entire small intestine, generating slow and prolonged release of glucose, according to the in vitro Englyst assay. The aim of this work was to conduct a meta-analysis of up-to-date evidence to evaluate the association between SDS consumption and a reduction in the postprandial glycemic response, including extended glycemic index (EGI) or glycemic profile (GP) parameters, during in vivo digestion. We searched the Web of Science, PubMed, Europe PMC, Cochrane Library, and Embase to identify related articles published up to September 2022. Human trials investigating the effect of the SDS amount on the postprandial glucose profile were estimated at the standard mean difference (SMD), with a 95% confidence interval (CI), using random effect models. The review followed the systematic reviews and meta-analyses (PRISMA) guidelines. The meta-analysis included a total of 65 participants. The results revealed that the EGI experienced a greater increase (SMD = 24.61, I2 = 79.2%, *p* < 0.01) after SDS intake, while the GP exhibited similar trends (SMD = 29.18, I2 = 73.3%, *p* < 0.01). High heterogeneity vanished in the subgroup and sensitivity analysis (EGI: I2 = 14.6%, *p* = 0.31; GP: I2 = 0.0%, *p* = 0.97). There was no evidence of publication bias for EGI (*p* = 0.41) or GP (*p* = 0.99).The present meta-analysis provides evidence that SDS intake is positively correlated with EGI and GP levels. The quantitative relationship of the reduction in the postprandial glycemic response and SDS consumption was used to quantify the slow digestion property on an extended time scale, and supplement the in vitro concept of SDS.

## 1. Introduction

Diabetes and its related chronic disease have already become major public health problems and more prominent social issues, with the number of new cases growing exponentially in recent years. According to the latest International Diabetes Federation (IDF) Diabetes Atlas in 2021 [1], the global estimate of adults (20–79 years) living with diabetes was 537 million, causing at least $ 966 billion dollars in health expenditure. Notably, 541 million adults are at an increased risk of developing type 2 diabetes, with the total number of people living with diabetes is projected to rise to 643 million by 2030, and 783 million by 2045. Compelling evidence from epidemiologic studies indicated that the impaired glucose tolerance and the impaired fasting glucose from ingested dietary starches form an intermediate stage in the history of Type 2 diabetes [2,3]. Current advances in food science and nutrition have concluded that slowing starch digestion rate helps to maintain postprandial blood glucose homeostasis and energy balance, for human benefits of well-being. 

Starch is the main glycemic carbohydrate in cereals, legumes, roots or tubers, and is one of the most important energy sources for humans. According to its physiological effects after consumption, starch has been classified into three consecutive nutritional fractions: rapidly digestible starch (RDS), slowly digestible starch (SDS), and resistant starch (RS), using the in vitro Englyst assay [4]. Among these, SDS as an intermediate starch fraction between RDS and RS, is digested between 20 and 120 min, resulting in the slow and sustained release of blood glucose, along with lower postprandial insulinaemia and circulating non-esterified fatty acids (NEFA), and stimulus-secretion of glucagon-like peptide-1 (GLP-1) and glucose-dependent insulinotropic polypeptide (GIP), participating in the gut–brain axis [5,6,7,8]. In recent decades, the glycemic index (GI) was proposed as a physiological classification of carbohydrate quality and a measure of how a carbohydrate-containing food raises blood glucose [3]. RDS is abundant in highly processed foods and is the most significant contributor to a rapid influx of glucose into the bloodstream, resulting in a high GI, whereas RS behaves like dietary fiber for fermentation in the colon, to release short-chain fatty acids for colonic health, with an accompanying low GI feature [3]. However, the GI concept has some limitations, since a low GI does not necessarily mean a slow and sustained release of blood glucose after SDS-rich food intake, especially for foods rich in non-starch filler including proteins, lipids, and gum [3,9]. 

The importance of delayed postprandial glycemia has received more attention in recent years, and it is imperative to develop an in vivo approach to assess the real SDS material for the prevention and intervention of metabolic syndromes, due to the inaccuracy of in vitro procedures and complexities of glucose metabolism. Based on the in vivo profile of SDS digestion, the extended glycemic index (EGI) has been proposed as a complementary method of the GI [10]. The value of the EGI corresponds to the area under the glycemic response curve over a prolonged time period, compared to the glucose control curve above or at the baseline, which could be used to quantify and evaluate the nutritional properties of SDS-rich foods. Recently, the glycemic profile (GP) has been developed to calculate the duration of the net glucose increment divided by the peak value of glucose [11]. Numerous earlier reviews have been carried out which focus only on starch digestibility or carbohydrate and glycemic response: however, little work has been reported on the relation of nutritional fraction of SDS and the reduction of glycemic responses. The aim of this work was to perform a meta-analysis of up-to-date evidence to evaluate the association between SDS intake and the EGI or GP during in vivo digestion. The preferred reporting items for systematic reviews and meta-analyses (PRISMA) guidelines were followed for this systematic review and meta-analysis [12].

## 2. Materials and Methods

### 2.1. Data Sources and Searches

The comprehensive literature searches up to September 2022 were conducted by two investigators, including all research evidence on the associations between starch intake and the postprandial glycemic response. The following online databases were searched for all studies published: Web of Science, PubMed, Europe PMC, Cochrane Library, and Embase. The following terms were used for the search within the databases: “slowly digestible starch”, “glucose”, “glycemic index”, “extended glycemic index”, and “glycemic profile”. The search strategy was as follows: “slowly digestible starch”; “slowly digestible starch” AND “glucose”; “slowly digestible starch” AND “glycemic index” AND “glycemic profile”; “slowly digestible starch” AND “glycemic index”; “slowly digestible starch” AND “extended glycemic index”; “slowly digestible starch” AND “glycemic profile”. Studies were limited to English. 

### 2.2. Study Selection

Two investigators independently assessed the titles, abstracts, and full texts from the literature searches to make decisions regarding potentially eligible articles. In case of any disagreement, a third researcher was involved in the decision-making process. The selection of studies was based on predetermined exclusion and inclusion criteria. Studies were included if (1) the dietary intervention was performed by SDS intake, (2) the exposure of interest was the EGI as measured by the GI, (3) full-text articles were accessible in English, and (4) the subjects were healthy humans. Studies that did not fulfill the defined inclusion criteria were excluded if (1) they did not apply SDS as an intervention, (2) they were randomized to multi-factorial interventions, while SDS intervention could not be extracted, (3) they did not measure the GI or EGI, and (4) they were review articles. 

### 2.3. Data Extraction and Quality Assessment

To extract relevant data from the included studies, the relevant information was extracted by two investigators: (1) first author’s name; (2) year of publication; (3) study design; (4) population and sex; (5) adjustment period; and (6) mean (standard deviation, SD) for the observed values of SDS, the EGI, and the GP. A third investigator resolved eventual disagreements by consensus. If data were missing, the authors were contacted to acquire additional information. A study was excluded where there was no response, denial of provision, or data loss. The Cochrane Collaboration’s tool was used for assessing the risk of bias of each study. The low, high, or unclear risk of bias in each of the 7 domains was rated.

### 2.4. Statistical Analysis

Meta-analysis was performed to estimate the EGI and GP impact of the SDS consumption. The results were presented with the standardized mean difference (SMD) and their 95% CI, using a random-effect model. Heterogeneity between studies was assessed via the I^2^ value and the *p*-value obtained from the Q-test, where values above 50% were defined as substantially heterogenous [13]. Results were considered statistically significant when *p* < 0.05.

In case of substantial heterogeneity, subgroup analysis by sex was employed to study the potential sources of heterogeneity. Sensitivity analysis was performed, excluding one study at a time, to clarify whether the results were simplified due to a large study or a study with extreme data. The sensitivity analysis reports had the most positive and negative influence on the summary estimate. The publication bias was statistically evaluated by Egger’s test, and the result was considered to indicate small study effects when *p* < 0.10. All analyses were carried out using the STATA 14.0 software (STATA, College Station, TX, USA).

## 3. Results

### 3.1. Research Data

The flow and selection of studies from our search strategy are summarized in Figure 1. A total of 4284 studies were identified as eligible in the database search. Of these, 2238 studies remained after removing duplicates. After screening, 13 studies remained and were retrieved in full text, to be assessed according to the set exclusion and inclusion criteria. Of these 13 studies, 8 studies were excluded due to not assessing the effect of SDS levels, being non-human studies, or insufficient information. Thus, a total of 5 studies were involved in a meta-analysis to reveal the association between SDS consumption and reduction of glycemic responses, including EGI and GP.

### 3.2. Study Characteristics and Quality Assessment

Table 1 presents the characteristics of the five studies [10,14,15,16,17]. Changes in the SDS content and the GI were reported in all studies. A total of 65 participants were selected for this meta-analysis, of whom 36 were female and 29 were male, with an age range of 18–40 years. Furthermore, these participants had no use of blood donation, antibiotics, or medication, in the past three months; gastrointestinal dysfunction or surgery; inflammatory disease; or diabetes. Among the interventions selected, one study was carried out in a parallel design, and four studies were carried out in a crossover design. In the interventions, all foods and drinks of the diet were supplied. Prior to dietary intervention, all subjects needed to adapt for a few days, and fast for 9–16 h.

Five studies used different modification methods of starch to increase the SDS content for the intervention. Two studies reported a significant reduction in blood glucose levels after consuming SDS, due to extended glucose release [10,15], and three studies reported a control and stabilization effect on levels of blood sugar, in the intervention group [14,16,17]. Even without reducing postprandial blood glucose levels, products containing slowly digestible starches have long-term beneficial effects. In addition, values of the EGI and GP were calculated by two investigators, according to their respective concept definitions.

Table 2 shows the risk of bias assessment in each study. All studies did not describe the method of allocation concealment. The methods of randomization in two studies were specified [14,15]. For the participants and personnel in most of the studies, a double-blind design was adopted. There were only 2 studies rated as low-risk for the blinding of outcome assessments [10,16], whereas the others were all rated as unclear risk. The incomplete outcome data was also observed in a similar situation. Except 2 studies [10,17], the rest of the studies on selective outcome reporting were rated as unclear risk. There were no other sources of bias identified for the included studies. 

### 3.3. Associations of SDS Intake with EGI or GP

The relationship between the SDS content and the EGI or GP was determined using a meta-analysis with the five selected studies. Figure 2 presents the results from the primary meta-analysis. This analysis revealed that either the EGI or GP was significantly dependent on SDS content (*p* < 0.01). The SMD generated by SDS intake was 24.61 for the EGI (95% CI, 15.71–33.52) and 29.18 for the GP (95% CI, 19.63–38.73). A pronounced positive association between the EGI or GP and SDS intake was found. In addition, moderate heterogeneity was observed in the two variables, with I^2^ values of 79.2% and 73.3% (Figure 2).

### 3.4. Subgroup and Sensitivity Analyses

In order to identify the heterogeneity, subgroup and sensitivity analyses were carried out, shown in Figure 3. As the number of included studies is limited, subgroup analysis could only be performed based on gender, in the feature table. The results of the subgroup analysis according to population sex are displayed in Table 2. Heterogeneity was reduced for the EGI and GP within the M subgroup (EGI: I^2^ = 10.5%, *p* = 0.31; GP: I^2^ = 37.0%, *p* = 0.03), while significant heterogeneity remained in the F subgroup (EGI: I^2^ = 86.4%, *p* < 0.01; GP: I^2^ = 77.9%, *p* < 0.01). These results indicate that there may be sources of heterogeneity other than sex. Furthermore, the positive associations between the EGI or GP and SDS intake did not change by the variable.

To further seek other possible sources of heterogeneity, an additional sensitivity analysis was conducted, where each study was excluded one at a time. Two works from Ren et al. [14] and Péronnet et al. [15] were responsible for the heterogeneity in the sensitivity analysis for EGI. When we withdraw the two studies, the heterogeneity dropped considerably (SMD = 17.23; 95% CI, 13.30–21.17; I^2^ = 14.6%, *p* = 0.31). For GP, high heterogeneity vanished when the works from Péronnet et al. [15] and Eelderink et al. [17] were excluded (SMD = 21.64; 95% CI, 16.86–26.43; I^2^ = 0.0%, *p* = 0.97). 

### 3.5. Publication Bias

The possibility of publication bias was explored by plotting the mean differences against the standard errors for the EGI or GP and SDS intake (Table 3). Based on Egger’s test (Table 4), no publication bias was presented for the EGI (*p* = 0.40) or GP (*p* = 0.99).

## 4. Discussion

There were 5 studies for analysis of the association of EGI or GP with SDS. In this meta-analysis, we observed that the consumption of SDS had a positive effect on the EGI and GP. A significant increase was observed in the study for either EGI or GP (EGI: SMD = 24.61; 95% CI, 15.71–33.52; GP: SMD = 29.18; 95% CI, 19.63–38.73). This phenomenon may be related to the digestibility of slowly digestible carbohydrates. 

SDS is not an inherent property of starch-based foods such as cereals, legumes, roots, and tubers, due to the SDS scarcity in traditional thermally processed foods. According to the in vitro digestion rate for quantification, SDS is an approach-oriented concept for the intermediate fraction between RDS and RS, without specification of structural requirements, which is different from the concept of GI. Raw normal maize starch granules are an ideal source of SDS, based on in vitro tests [3], but is rapidly and completely digested in vivo, in the upper portion of the small intestine [18]. For enzyme-catalyzed digestion, there are two fundamental structural bases and mechanisms for SDS: (a) starch structural features limiting the rate of enzymatic action and (b) physical barriers that prevent access/binding of enzymes. Current available literature has shown that starch products rich in SDS can be obtained using physical, enzymatic, and chemical, as well as genetic modifications [19,20,21]. For instance, physical processing approaches provide a good opportunity to produce SDS-state starch through the controlled gelatinization, hydrothermal treatment, recrystallization, and food matrix interactions [3]. Enzymatic modification is a promising approach to restructure the fine structural pattern of amylopectin, including the branched chain length and branch density, leading to a higher proportion of SDS [20]. For the chemical modification of starch, the hydroxyl substitution, glucan chain cross-linking or acid degradation of granular structures results in tailor-made low digestibility [15]. Genetic modification is a promising strategy to change the enzyme activities of starch biosynthesis and generate several new cultivars with desired functionality [3]. To date, several commercial products with high SDS content have been developed, such as EDP^®^ plain biscuits (Danone), BelVita^®^ breakfast biscuits (Mondelez), Cluster Dextrin™ highly branched cyclic dextrin (Ezaki Glico), and SUSTRA™ 2434 slowly digestible carbohydrate (Ingredion). In 2011, the European Food Safety Authority (EFSA) also validated that carbohydrate-rich foods containing more than 55% of the available carbohydrates from starch, at least 40% of which is SDS, are considered high-SDS foods [22]. This evidence further reveals that the SDS amount in food was the ultimate standard to quantify the health effect. In this study, two sets of data also showed similar data, with an SDS of 52.0% and 40.5%, respectively [14,17].

Glycemic starch is enzymatically-degraded in the upper gastrointestinal tract, with the cooperation of amylolytic α-amylases and brush border glucogenic enzymes. After dietary consumption, the glucose polymer is first cleaved with α-amylases to α-limit dextrins as well as oligosaccharides, which are further digested using the small intestinal brush border enzymes of maltase-glucoamylase and sucrase-isomaltase into glucose, for absorption [23]. SDS is slowly digested throughout the entire length of the small intestine, providing sustained glucose release with a low initial rise in blood glucose levels, and a subsequently slow and prolonged release of glucose, similar to the health benefits of human well-being for low-GI foods. However, there is no existing in vivo assay to quantify the amount of physiology-oriented SDS (SDS), and a low GI food does not necessarily mean a slow and sustained release of glucose in the small intestine. It is amazing that the potential health benefits of SDS are inferred from low-GI foods, and the exact consequence is still under investigation. 

After the digestion process of SDS, glucose is transported to the wall of the small intestine and then into the portal vein by active transport, via a sodium-dependent transporter, for the assimilation. Incretin hormones (CCK, GIP, GLP-1, and PYY) are triggered from endocrine cells by distal glucose stimuli in the digestive tract and decrease the gastric emptying rate related to satiety and food intake via ileal braking activation, which can be associated with glycaemic control and weight loss [24]. For SDS intake in the gastrointestinal tract, a slow glucose release property accompanied by a low insulin level was observed, which might provide wide health benefits to reduce the risk of diabetes or metabolic syndrome [25]. Consuming SDS-rich cereal products is associated with a slower rate of appearance (RaE) kinetics, and its consumption has a 15-fold lower RaE than that of low SDS food, resulting in a low glycemic response [26]. Consequently, starchy foods rich in SDS play a more outstanding role in the RaE regulation. Recently, a statistical meta-analysis of randomized controlled trials first proposed the relationship between the SDS amount in cereal foods and blood RaE [27]. SDS-rich foods contribute to a lower RaE at the end of the postprandial period than that of cereal products lacking in SDS. Regulating the postprandial glucose availability from breakfast would reduce the plasma exogenous glucose appearance and alter the glycemic control for the following lunch [15]. SDS intake also extended exogenous glucose oxidation, induced a sustained elevation of GLP-1 and GIP, and generated lower levels of plasma C-peptide, circulating NEFAs, and triacylglycerol-rich lipoproteins [28,29,30,31]. In addition to dietary intervention for chronic metabolic syndromes, high SDS food with a moderate postprandial glycemic response has implications for physical performance, mental performance, and obesity control. 

Recently, the glycemic glucose fraction was classified into rapidly available glucose (RAG) and slowly available glucose (SAG) that takes into account the digestion and absorption rates of glucose, and RAG was positively correlated with GI [32]. High SDS foods generated the prolonged release of blood glucose profiles with remarkably lower values at the initial 30–60 min, as well as remarkably higher values at 120–240 min, displaying the sustained and balanced energy release [3,33]. EGI or GP is considered an in vivo approach for the evaluation of the physiological quality of SDS, complementary to the concept of the GI, as well as SDS. Our meta-analysis quantitatively analyzed the associations of the EGI and GP with the consumption of SDS in the maintenance of glucose homeostasis. Both the EGI and GP increased with increasing SDS intake, which provided a better understanding of the amount and physiological nature of SDS, from the viewpoint of in vivo studies.

## 5. Conclusions

This meta-analysis provides evidence that SDS intake increased the value of EGI and GP, but to different extents. The quantitative relation of the reduction in the postprandial glycemic response with SDS consumption was used to quantify the slow digestion property on an extended time scale, and supplemented the in vitro concept of SDS. Thus, considering both the structure of carbohydrates and the physiological process of carbohydrate digestion can promote the exploitation of novel dietary approaches and products to achieve expected slow carbohydrate bioavailability for enhanced health. Potential limitations of our meta-analyses should be considered when interpreting the findings. First, heterogeneity was moderate for two variables. Sensitivity analysis suggested the impact of some of the individual studies on two variables, although the SMD results remained nearly similar after their omission. For the EGI, the selected studies from Ren et al. [13] and Péronnet et al. [14] were withdrawn, and the heterogeneity dropped considerably. This may be explained by the lowest levels of EGI and GP for the two studies. For GP, high heterogeneity vanished when the studies reported by Péronnet et al. [14] and Eelderink et al. [16] were excluded. It may be ascribed to the content of SDS. Obviously, compared with other studies, the subjects in the two studies consumed the highest and lowest SDS content, respectively. Second, the different included studies had a wide range of variability in source, preparation of SDS, and subjects, which could have confused our findings. Ideally, we were supposed to adopt subgroup analysis of multiple factors to address this problem. However, resulting from the limitation of extracting sufficient data from each study, we only performed subgroup analysis of population sex. Notably, for the studies of Péronnet et al. [14] and Eelderink et al. [16], only female or male subjects were recruited to participate in the study, thereby resulting in the heterogeneity of the meta-analysis. A further study is currently underway to unlock the relationship between the SDS amount and the appropriate glycemic response, and to conduct SDS intervention studies in human beings with an emphasis on EGI or GP, which will help to design tailor-made food formulations for dietary intervention.

## Figures and Tables

**Figure 1 foods-12-00089-f001:**
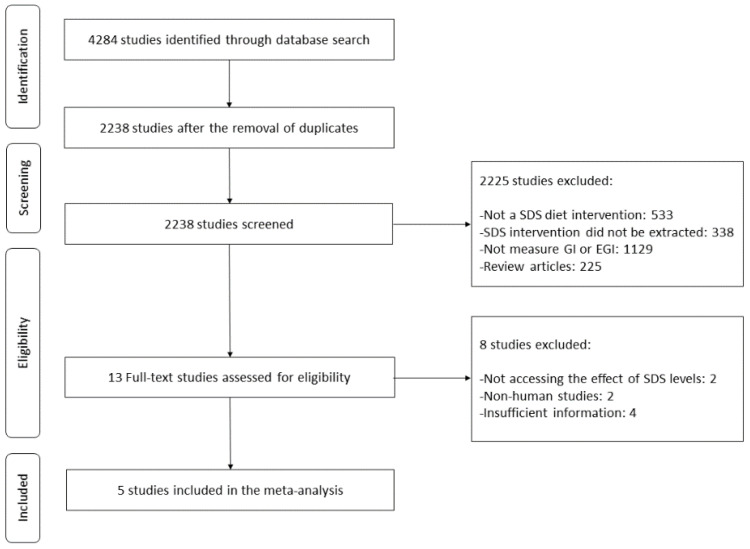
Flow chart of study selection for the meta-analysis.

**Figure 2 foods-12-00089-f002:**
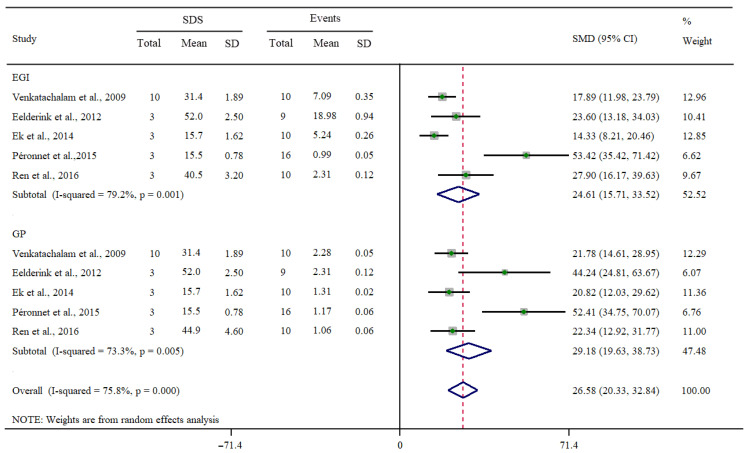
Forest plots of the effect of SDS intake on EGI and GP (D + L: random effect model) [10,14,15,16,17]. Each data was presented as weight (%), as well as SMD and 95% CI.

**Figure 3 foods-12-00089-f003:**
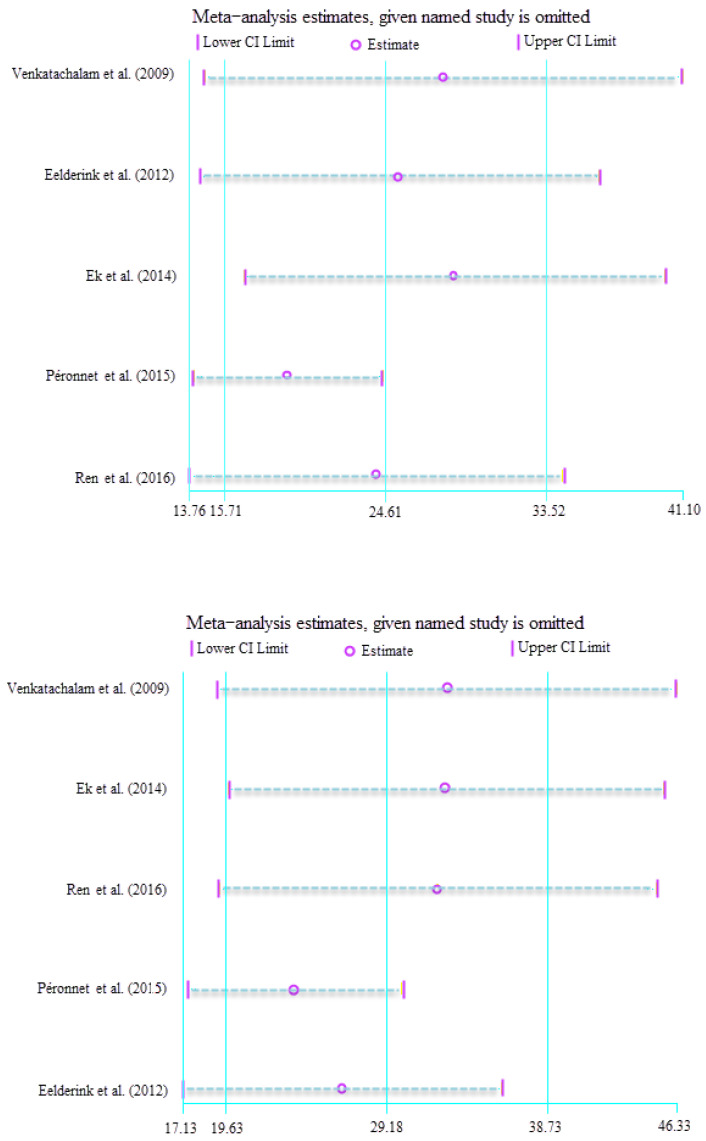
Sensitivity analysis of SDS intake and EGI (**top**), GP (**bottom**) [10,14,15,16,17].

**Table 1 foods-12-00089-t001:** Characteristics of the five studies included in the individual analysis.

Study (Year)	Study Design	Age (Mean ± SD)	BMI (Mean ± SD)	Geographic location	Population	SDS Content (Mean ± SD)	Adjustment Period	Index (Mean ± SD)	
								EGI	GP
Venkatachalam et al. (2009) [10]	crossover	21 ± 0.5	23 ± 0.6	North America	Human (7M, 3F)	31.4 ± 1.89	fasted for >9 h	7.09 ± 0.35	2.28 ± 0.05
Eelderink et al. (2012) [17]	crossover	27 ± 5.1	22.3 ± 2.0	Europe	Human (9M)	52.0 ± 2.50	fasted overnight, limited physical activity	18.98 ± 0.94	2.31 ± 0.12
Ek et al. (2014) [16]	crossover	26	20.8	Oceania	Human (12M, 15F)	15.7 ± 1.62	fasted for 10–12 h, physical activity	5.24 ± 0.26	1.31 ± 0.02
Péronnet et al. (2015) [15]	crossover	18–40	20–25	North America	Human (16F)	15.5 ± 0.78	energy intake, physical activity, taking an oral contraceptive	0.99 ± 0.05	1.17 ± 0.06
Ren et al. (2016) [14]	crossover	23 ± 4.0	26.15 ± 2.6	Asia	Human (3M, 7F)	40.5 ± 3.20	fasted for 10–12 h	2.31 ± 0.12	1.06 ± 0.06

F, female; M, male; Lab, laboratory; SD, standard deviation.

**Table 2 foods-12-00089-t002:** Assessment of risk of bias in the included studies using Cochrane criteria.

Study	Random Sequence Generation	Allocation Concealment	Blinding of Participants & Personnel	Blinding of Outcome Assessments	Incomplete Outcome Data	Selective Reporting	Other Bias
Venkatachalam et al. (2009) [10]	Un	Un	L	L	L	L	L
Eelderink et al. (2012) [17]	Un	Un	L	Un	Un	L	L
Ek et al. (2014) [16]	Un	Un	L	L	L	Un	L
Péronnet et al. (2015) [15]	L	Un	L	Un	Un	Un	L
Ren et al. (2016) [14]	L	Un	Un	Un	L	Un	L

L: low risk of bias; H: high risk of bias; Un: unclear risk of bias.

**Table 3 foods-12-00089-t003:** Subgroup analyses of SDS intake and EGI, GP.

	EGI			GP		
	SMD (95% CI)	P*_H_*	I^2^, %	SMD (95% CI)	P*_H_*	I^2^, %
Total	24.01 (21.72, 26.29)	<0.01	78.6	28.57 (26.18, 30.96)	<0.01	69.5
M	17.57 (16.16, 18.98)	0.31	10.5	23.98 (21.66, 26.31)	0.03	37.0
F	31.40 (27.07, 35.74)	<0.01	86.4	32.31 (28.35, 36.26)	<0.01	77.9

F, female; M, male; PH, P-Heterogeneity.

**Table 4 foods-12-00089-t004:** Egger’s test of EGI, GP, and SDS intake.

Group	Std. Eff.	Coef.	Std. Err.	t	P > |t|	[95% CI]
EGI	slope	6.56249	7.982013	0.82	0.56	−94.86, 107.98
	bias	3.14865	2.29361	1.37	0.40	−25.99, 32.29
GP	slope	21.53624	4.868787	4.42	0.14	−40.33, 83.40
	bias	0.02540	1.150894	0.02	0.99	−14.60, 14.65

## Data Availability

The data presented in this study are available in the review.

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
