# Peer review of "Association of Slowly Digestible Starch Intake with Reduction of Postprandial Glycemic Response: An Update Meta-Analysis"

_foods, 2022, doi:10.3390/foods12010089_

Round 1
Reviewer 1 Report
Dear Editor and Authors,
The manuscript is very well written and deals with sought-after and important knowledge related to human nutrition. The authors made a very good review of the existing knowledge, it is a pity that by September 2021, is it possible to at least mention something about newer reports in the conclusion. In the introduction, it would be worth mentioning what reviews have been carried out so far on starch digestibility and in order to emphasize what is new in this current work. As for the conducted review, i.e. the selection and method of excluding available sources, it can be seen that it was carried out very substantively and without objections. The statistical analysis and description of the results were well done.
Best regards,
Author Response
The manuscript is very well written and deals with sought-after and important knowledge related to human nutrition. The authors made a very good review of the existing knowledge, it is a pity that by September 2021, is it possible to at least mention something about newer reports in the conclusion. In the introduction, it would be worth mentioning what reviews have been carried out so far on starch digestibility and in order to emphasize what is new in this current work. As for the conducted review, i.e. the selection and method of excluding available sources, it can be seen that it was carried out very substantively and without objections. The statistical analysis and description of the results were well done.
RESPONSE: The recommendation is highly appreciated, and we have taken it up. Please, see the whole revised manuscript.
Reviewer 2 Report
Wang et al., conducted a meta-analysis of SDS intake and its association with EGI or GP. This meta-analysis study was planned and performed very well. Surprisingly, there were only 5 studies that fully fit this study's criteria.
Hence it indicates that more future work needs in SDS intervention studies in human beings with an emphasis on EGI or GP.
It is with minor typo corrections for In-vitro and in-vivo, which should be in italic (Abstract).
Author Response
Wang et al., conducted a meta-analysis of SDS intake and its association with EGI or GP. This meta-analysis study was planned and performed very well. Surprisingly, there were only 5 studies that fully fit this study's criteria.
RESPONSE: Some research paper focus only on in vitro starch digestibility, or in vivo glycemic response. Also, numerous earlier reviews have been carried out to focus only on starch digestibility or carbohydrate and glycemic response. An attempt has been made to select the related paper to conducted a meta-analysis of relation of SDS intake and EGI or GP
Hence it indicates that more future work needs in SDS intervention studies in human beings with an emphasis on EGI or GP.
RESPONSE: The manuscript was revised accordingly to comment. Please, see the revised manuscript.
It is with minor typo corrections for In-vitro and in-vivo, which should be in italic (Abstract).
RESPONSE: Revised.
Reviewer 3 Report
1. What is the novelty and significance of this review on intake of slow digestible starch and its postprandial glycemic response, especially when there are already many number of similar reviews are easily available.
2. Some of the investigated sample literatures are more than five years old. They dont represent the state of art improvements in the field of study.
Even though the manuscript is well written, Authors should include their response for above mentioned factors and should focus on recent trends in the slow digestible starch to improve the standard of this review article.
Author Response
- What is the novelty and significance of this review on intake of slow digestible starch and its postprandial glycemic response, especially when there are already many number of similar reviews are easily available.
RESPONSE: An attempt has been made to explain the novelty and significance of this review. Please, see the revised manuscript.
- Some of the investigated sample literatures are more than five years old. They dont represent the state of art improvements in the field of study.
RESPONSE: Some research paper focus only on in vitro starch digestibility, or in vivo glycemic response. Also, numerous earlier reviews have been carried out to focus only on starch digestibility or carbohydrate and glycemic response. An attempt has been made to select the related paper to conducted a meta-analysis of relation of SDS intake and EGI or GP
Even though the manuscript is well written, Authors should include their response for above mentioned factors and should focus on recent trends in the slow digestible starch to improve the standard of this review article.
RESPONSE: Recommendations accepted. Please, see the revised manuscript.
Reviewer 4 Report
Please see the attached file

Author Response
Comment for authors
The authors report : Slowly digestible starch intake is associated with reduction of 3 postprandial glycemic response: an update meta-analysis
The English language is generally good, but some corrections are suggested below :
RESPONSE: The manuscript was revised accordingly to comment. Please, see the revised manuscript.
Title : I suggest you change the title to: association of slowly digestible starch intake with reduction of 3 postprandial glycemic response : an update meta-analysis
RESPONSE: Revised.
Line 6 : use the same writing character for 1800 Lihu
RESPONSE: Revised.
Abstract
Line 14,17, and 29: in vivo and in vitro are written in italics, correct throughout the manuscript
RESPONSE: Revised.
Line 31 : delete introduction
RESPONSE: Revised.
Introduction
Line 35 : replace worldwide with another term to avoid repetition because you gave it before
RESPONSE: Revised.
Give briefly at the end of the introduction the procedure to follow in order to carry out this work
RESPONSE: The recommendation is highly appreciated, and we have taken it up. Please, see the revised manuscript.
Results
Figure 1 : Try to make the writing character bigger so that the figure is more visible.
RESPONSE: Revised.
Table 1 : Correct Ren, X., et al. to Ren et al. ; Arrange the writing of Age (Mean ± SD)
RESPONSE: Revised.
Table 2 : Correct also Ren, X., et al. to Ren et al. ; Arrange the writing of the first line
RESPONSE: Revised.
Indicate figure 3 in the text
RESPONSE: Revised.
Table 3 : Arrange the writing of Total
RESPONSE: Revised.
Indicate Table 4 in the text
RESPONSE: Revised.
Discussion
Line 249 : replace data by another term to avoid repetition
RESPONSE: Revised.
It is better to put the limitations in the conclusion part.
RESPONSE: Recommendations accepted. Please, see the revised manuscript.
Round 2
Reviewer 3 Report
The authors have addressed the points raised by the reviewer. Thus the manuscript may be considered for publication in foods.